# Promoting a Healthy Lifestyle through Mindfulness in University Students: A Randomized Controlled Trial

**DOI:** 10.3390/nu12082450

**Published:** 2020-08-14

**Authors:** Encarnación Soriano-Ayala, Alberto Amutio, Clemente Franco, Israel Mañas

**Affiliations:** 1Department of Education, Universidad de Almería, 04120 Almeria, Spain; esoriano@ual.es; 2Department of Work Relations and Social Work, University of the Basque Country (UPV/EHU), 48940 Bilbao, Spain; 3Faculty of Education and Social Sciences, Universidad Andres Bello, Santiago 7591538, Chile; 4Department of Psychology, Universidad de Almería, 04120 Almeria, Spain; cfranco@ual.es; 5Department of Psychology, Universitat Oberta de Catalunya, 08018 Barcelona, Spain; imanas@ual.es; 6Hum-760 Research Team, Health Research Centre, University of Almería, 04120 Almeria, Spain

**Keywords:** life habits, mindfulness, flow meditation, university students, controlled trial

## Abstract

The present study explored the effects of a second-generation mindfulness-based intervention known as *flow meditation* (*Meditación-Fluir*) in the improvement of healthy life behaviors. A sample of university students (*n* = 51) in Spain were randomly assigned to a seven-week mindfulness treatment or a waiting list control group. Results showed that compared to the control group, individuals in the mindfulness group demonstrated significant improvements across all outcome measures including healthy eating habits (balanced diet, intake rate, snacking between meals, decrease in consumption by negative emotional states, increased consumption by negative emotional states, amount of consumption, meal times, consumption of low-fat products), tobacco, alcohol, and cannabis consumption, and resting habits. There were differences between males and females in some of these variables and a better effect of the treatment was evident in the females of the experimental group when compared to the males. The *flow meditation* program shows promise for fostering a healthy lifestyle, thus decreasing behaviors related to maladaptive eating, tobacco, alcohol, and cannabis consumption as well as negative rest habits in university students. This mindfulness program could significantly contribute to the treatment of eating disorders and addictions, wherein negative emotional states and impulsivity are central features of the condition.

## 1. Introduction

University students may be a population at biological and psychosocial risk, often with maladaptive health behaviors and life habits including unhealthy eating, inadequate rest habits, and substance abuse (e.g., tobacco, alcohol, cannabis). In Spain, these students are distancing themselves from the traditional Mediterranean diet and food patterns [1]. Lifestyle changes of the past few decades have increasingly led to changes in eating habits such as the reduced consumption of fresh or minimally processed foods and increased consumption of ultra-processed products [2]. New eating habits are reflected in an unhealthy diet and various health-related problems including being overweight [1,3].

Substance abuse among young people is also an issue of great concern for their long-term psychological and physical health effects [4]. For instance, acute adverse health effects following cannabis use were reported by 40% of the participants in a study conducted by Contreras et al. [5]. According to this study the most common effects were paranoia, increased heart rate, panic attacks, or anxiety, whereas fewer reported experiencing more severe effects like hallucinations or loss of consciousness.

For its part, tobacco smoking provokes respiratory and cardiovascular diseases and multiple types of cancer. Research has revealed a direct association between having externalizing problems (e.g., aggressive behaviors) and the initiation and continuation of cigarette smoking [6]. Thus, youths scoring high in externalizing are more likely to initiate cigarette smoking. In this sense, controlling external variables should be a higher priority. Similarly, alcohol consumption is subject to social and environmental factors (e.g., peer pressure) as well as cognitive factors (e.g., impulsivity, positive expectations and sensation seeking, anxiety sensitivity, and hopelessness), among others [7,8]. In addition, alcohol consumption among university students is related to the appearance of problematic behaviors and violence as well as decreased academic performance.

Poor sleep quality is also a recurrent feature of student life and affects not only cognitive processes, but also recovery from stress and the elimination of fatigue, increasing the possibility of poor school performance [9]. Moreover, several studies have shown that sleeping problems including bad rest habits, correlate positively with both uncontrolled and emotional eating and substance use in differing populations [4,10,11,12]. In contrast, sleep duration and sleep quality have been associated with healthy diets [2].

All these maladaptive behaviors in students can be related to the high prevalence of stress in this population, emotional dysregulation with lack of self-control, and identity issues, [7,10,12]. The modification of these unhealthy behaviors is a potential target for intervention as well as for improving school success. Recommended treatments are psychological and behavioral counselling, different multimodal therapeutic programs, psychotropic medication, and stress management techniques including mindfulness [12,13,14]. Likewise, positive personal and family-domestic habits such as looking for calm and tranquil sensations and eating as a family with the TV turned off are related to the capacity of controlling repetitive patterns of thoughts and behaviors, and improved well-being and academic performance [15].

### 1.1. Mindfulness-Based Meditation

The practice of mindfulness-based meditation has been tested as a potential therapeutic intervention for a wide range of clinical problems including eating disorders and other addictions [7,14,16] wherein anxiety and impulsivity are key features [17]. The value of these strategies has been widely recognized for a wide variety of psychological and psychosomatic disorders associated with emotional dysregulation [18,19]. More specifically, different mindfulness-based approaches have been used including Dialectical Behavior Therapy (DBT, [20,21,22,23]), Acceptance and Commitment Therapy (ACT, [24]), Mindfulness-Based Stress Reduction (MBSR, [25]); Mindfulness-Based Cognitive Therapy (MBCT, [26]), and Mindfulness-Based Eating Awareness Training program (MB-EAT, Kristeller & Wolever, 2014, [16,27]). Mindfulness training should target specific behavioral impulses that an individual has difficulty controlling and also provide psycho-education. Attitudes regarding healthy eating and unhealthy behaviors are a key part of most psychological models that aim to foster positive habits.

According to Kabat-Zinn [25], mindfulness is to pay attention in a particular way, on purpose, in the present moment, and non-judgmentally. Furthermore, mindfulness meditation implies observing the thoughts and emotional reactions that occur at each moment by distancing from them (decentering) and not reacting before their presence in the automatic usual way thus, breaking the thinking–feeling–acting typical pattern [26].

### 1.2. The Present Study

There is growing literature suggesting that university students may be a population at risk due to the high rate of unhealthy behaviors that they present [28]. In fact, at least half of this population exhibits inadequate health behaviors that include poor eating and sleeping habits as well as alcohol and substance abuse [1]. Despite the large number of studies relating to the variables dealt with in this article, there is a shortage of studies in unhealthy behaviors carried out amongst university students, most of them being about eating habits and alcohol consumption, and there have been no previous studies analyzing them all simultaneously. To the authors’ knowledge, this is one of the first studies to specifically investigate the effects of mindfulness on a wide range of variables related to healthy lifestyle in university students. This is an important issue given the existing risk that maladaptive behaviors generate negative habits and disorders in later adulthood (e.g., eating disorders, alcoholism, drug addiction). Thus, the current study provides an experimental test of the effects of meditation practice in behaviors related to a healthy lifestyle including healthy eating, substance abuse control, and adequate rest habits.

Mindfulness has been shown to improve unhealthy life habits and several disorders in various clinical and non-clinical conditions [12,29,30,31]. Given the role of emotional problems and external factors in lifestyle patterns including sleep, food intake, and alcohol and drug use among youths, the goal of this study was to evaluate the effectiveness of a mindfulness-based intervention program known as *flow meditation* (*Meditación-Fluir*, FM, [32,33,34,35]. Specifically, this study evaluated the effects of mindfulness on a range of different life behaviors and variables including eating patterns (i.e., number of meal times, balanced diet, intake rate, consumption of low-fat products, snacking between meals, decrease in consumption due to negative emotional states (NES), and increase in consumption due to NES), tobacco, cannabis, alcohol consumption, and resting habits. Thus, it was hypothesized that compared to a waiting-list control group, students who completed the mindfulness intervention would demonstrate significant improvements in the aforementioned variables.

## 2. Materials and Methods

A randomized controlled trial was conducted to investigate the effects of mindfulness training on different variables related to lifestyle behaviors in a sample of university students. Assessments were taken at pre-test, post-test, and the intervention group was compared with a waiting list control group. Informed and written consent was obtained from all individual participants included in the study. All procedures performed in studies involving human participants were in accordance with the ethical standards of the institutional research committee and with the 1964 Helsinki Declaration and its later amendments or comparable ethical standards. This study was approved by the Bioethics Committee of the University of Almería-UALBIO2019/001-(Spain).

### 2.1. Participants

All participants were recruited from the University of Almeria (Spain) and were informed that they would be undertaking a program of training in mindfulness. A total of 51 eligible individuals (62.7% women) were recruited into the study and random assignment was employed to allocate participants to the intervention (*n* = 26) or control group (*n* = 25), mean age 20.94, SD = 18.31. More specifically, ballots, concealing the numbers 1 (control group) or 2 (intervention group) were placed in an equal number into an urn and each participant extracted one ballot from it. Randomization was conducted by a researcher not participating in the study and participants completed baseline assessments (pre-test) prior to allocation. Due to a logistical constraint, a sample size calculation was not conducted as sample size was governed by the maximum number of eligible participants that were willing to participate in the study at the time of enrolment.

The ANOVA and the Mann–Whitney U test for independent samples did not reveal statistically significant differences in the scores obtained before the treatment (pre-test; *p* < 0.05) by the participants of the control group and the experimental group, which means that the subjects of the two groups were extracted from the same population, so if statistically significant differences were found after the treatment (post-test), these would be due to the intervention. In addition, the Mann–Whitney U test for age was performed to compare the experimental and control groups, since they did not fit a normal distribution (Z = −1.40). There was no difference in age between the experimental and control group (*p* = 0.160). For the sex variable, the Chi-square had a *p* < 0.05, therefore there was also no difference in sex between both groups.

### 2.2. Procedure

A mindfulness training program called *flow meditation* (*FM*, Franco, 2009) [32] was administered to the intervention group using two-hour weekly group sessions for a period of seven weeks. The training program included (i) mindfulness exercises from Kabat-Zinns’ [36] stress-reduction program; (ii) mindfulness techniques used in acceptance and commitment therapy [37,38,39]; and (iii) exposure to and debate on metaphors and exercises used in Zen [40] and Vipassana meditation [41], which promotes values such as acceptance, forgiveness, and non-identification with mental events. Finally, some exercises that could be placed within the framework of logotherapy were also included [42], since their objective was to learn to overcome the existential void that arises from the lack of meaning in what we do and in the life we lead, and to take responsibility in this way for our own life. Logotherapy has demonstrated to be effective in the treatment of different problems including addictions, depression, anger, anxiety, and for fostering psychological well-being [43,44].

The effectiveness of *flow meditation* has been tested in various treatment studies with acceptable size effects (Amutio et al. 2016, 2018; Franco et al. 2016) [32,33,34,35,45]. The program includes existential, ethical, and spiritual aspects and values, and thus conforms to Van Gordons´ criteria for second generation mindfulness-based interventions (SG-MBIs) [46]. The program was facilitated by an instructor with extensive experience in both the practice and teaching of mindfulness meditation techniques.

The main purpose of *flow meditation* is for participants to learn to allow their thoughts to flow, without trying to modify them or interfere with them. The intervention is not concerned with teaching participants not to think about anything, but seeks to offer an alternative to conditioned automatic ways of reacting to inner and outer experiences. Each weekly session is structured as follows (Table 1):Discussion and feedback on the mindfulness meditation exercises practiced during the previous week.Ten-minute guided-mindful body-scan.Presentation of the various metaphors and exercises corresponding to each session.Practice of mindful breathing for 30 min.

Participants were requested to undertake the body-scan and mindful breathing exercises on a daily basis at home for 10 min and 30 min, respectively.

### 2.3. Ethical Considerations

All participants provided informed consent and the study was approved by the Bioethics Committee of the University of Almeria, Spain. The registered data for each of the instruments were alphanumerically coded, ensuring confidentiality and anonymity, in order to comply with the Personal Data Protection Act by the Ethics Committee for Research related to Human Beings (CEISH). International guidelines for studies with human subjects described in the Nuremberg Code and in the Declaration of Helsinki were applied. After completion of all assessment phases, the mindfulness training course was offered to control group participants.

### 2.4. Measures

The following measures were administered pre-and post-intervention:

Healthy Lifestyle Questionnaire. This 12-item measure evaluates the following factors of healthy lifestyle habits: eating habits (balanced diet and respect for meal times), tobacco consumption (e.g., “I feel well when I smoke”, and rest habits (e.g., “Usually I sleep around 7 or 8 h each night”). In the present study, six additional items were also introduced with the intention of evaluating the consumption of cannabis (three items), and the consumption of alcohol (three items), using for this purpose the same items that evaluate tobacco consumption, but changing the word tobacco for cannabis and alcohol, respectively, valued on a frequency scale of 1 (totally disagree) to 5 points (totally agree). The questionnaire has been validated in a Spanish population by Leyton et al. [47]. Cronbach’s alpha for this questionnaire was 0.79.

Lifestyle Questionnaire [48]. This 27-item questionnaire measures the following variables: Externality (eight items: e.g., “eating when you see advertisements on TV or magazine”); Decrease in Food Consumption due to NES (five items: e.g., “eating less or just not eating because you feel nervous”); Increase in Food Consumption due to NES (five items: e.g., “eating because you feel bored”); Quantity Consumption Pattern (four items: e.g., “filling your plate too much”); and Intake Rate (three items: e.g., “eating fast”), valued on a frequency scale of 1 (“always”)to 7 (“never”) points. The items “Snacking between meals” (whose frequency has been rated on a 5-point scale, from “never” to “everyday”) and “Consumption of low-fat meals and/or beverages” (whose frequency has been rated on a 7-point scale from “never” to “everyday”) are individually analyzed. Cronbach’s alpha for this questionnaire was 0.84.

### 2.5. Data Analyses

A pre-test/post-test design was used to explore the effectiveness of the intervention in the participants. The results in the applied instruments were considered dependent variables. First, descriptive statistics for the studied variables were calculated. A repeated measures ANOVA was performed when the data distributions were adjusted to a normal one and the homogeneity tests and the Box M test allowed it. When the statistical requirements to use this test were not met, we used the Wilcoxon test for intra-subject differences and the Mann–Whitney non-parametric U test for inter-subject differences. As a measure of the effect size, we used the partial *Eta* square for the repeated measures ANOVA and for the non-parametric tests, the Rosenthal *r* recommended by Wasserstein and Lazar [49] and is expressed as follows: *r* = *z*/√*N*. The results of *r* were compared by applying the Cohen [50] classification, which indicates that between 0.2 and 0.5, the effect size will be small; it will be moderate between 0.5 and 0.8; and above 0.8, it will be large. For this partial Eta square, around 0.01, little effect was considered; a moderate effect around 0.06; and a large effect at 0.14. All statistical analyses were conducted using SPSS version 25.0 (SPSS Inc., Chicago, IL, USA).

## 3. Results

The mean scores and standard deviations of the variables corresponding to the control and experimental groups in each of the study phases are shown in Table 2.

The ANOVA and the Wilcoxon test for related samples showed statistically significant differences between the scores of the pre-test and the post-test of the experimental group in all the variables studied, except with respect to meal times and the consumption of low-fat products. In the control group, significant differences were also observed in cannabis consumption; decrease in food consumption due to NES; increase in food consumption due to NES; and quantity of consumption, snacks between meals, consumption of low-fat products, and rate intake. However, in the experimental group, the mean scores between the pre-test and post-test increased in balanced diet and rest habits, and decreased in most of the remaining variables, thus showing the efficacy of the treatment carried out in this group. In the control group, differences between the mean scores between the pre-test and post-test indicated a worsening of healthy behaviors in these subjects (Table 2). In addition, in the experimental group, sizeable effects were obtained on balanced diet and rest habits (Figure 1 and Figure 2), whereas moderate effects were observed on alcohol consumption, respect for meal times, externality, decreased consumption due to NES, increased consumption due to NES, and quantity of consumption.

The ANOVA and the Mann-Whitney U test for independent samples did not reveal significant differences between men and women for the pre-test scores in the control and experimental groups. However, the ANOVA and the Wilcoxon test for related samples showed statistically significant differences between the pre-test and post-test scores for males in the experimental group in the following variables: alcohol consumption (Z = 2.12, *p* < 0.05), rest habits (F = 5.4, *p* < 0.05), externality (Z = −2.68, *p* < 0.01), decrease in consumption due to NES (Z = −2.21, *p* < 0.05), increase in consumption due to NES (Z = −2.37, *p* < 0.05), and amount of consumption (Z = −2.53, *p* < 0.05). Similarly, for females in the experimental group, significant differences were observed between the pre-test and post-test in all variables except meal times and the consumption of low-fat products. A better effect of the treatment was evident in the females of the experimental group than in males.

When analyzing the control group, significant differences can be observed in males between the pre-test and post-test in the variables quantity of consumption (Z = −2.41, *p* < 0.05), consumption of low-fat products (F = 4.6, *p* < 0.05), and rate of ingestion (Z = −2.12, *p* < 0.05), while in women, differences in cannabis consumption (Z = −2.00, *p* < 0.05), decrease in consumption due to NES (Z = −2.33, *p* < 0.05), increase in consumption due to NES (Z = −2.07, *p* < 0.05), and quantity of consumption (Z = −2.01, *p* < 0.05) were observed. However, whereas in the experimental group the differences were due to the increase in good health habits, in the control group, healthy habits decreased in the intervention period.

In relation to gender differences in the intervention group, males obtained large-size effects in rest habits (*ɳ*^2^*p* = 0.11) and moderate-size effects were observed in externality (*r* = 0.616), decreased consumption due to NES (*r* = 0.508), increased consumption due to NES (*r* = 0.545), and quantity of consumption (*r* = 0.582). In females, large effects were obtained in the balanced diet (*ɳ*^2^*p* = 0.148) and moderate effects were observed in tobacco consumption (*r* = 0.504), alcohol consumption *(r* = 0.535), rest habits (*ɳ*^2^*p =* 0.095), externality *(r* = 0.549), decrease in consumption due to NES (*r* = 0.586), increase in consumption due to NES (*r*= 0.606), amount of consumption (*r* = 0.607), and consumption of low-fat products (*ɳ*^2^*p* = 0.054).

## 4. Discussion

The current study compared a second-generation mindfulness-based intervention (SG-MBI; Van Gordon et al. [46,51]) known as *flow meditation* (FM, *Meditación-Fluir*) with a waiting list control group to investigate the effects of mindfulness on different behaviors leading to a healthy lifestyle in a sample of university students. Compared to first-generation mindfulness-based interventions, SG-MBIs employ a slightly different model of mindfulness that emphasizes the importance of non-attachment to self as well as to psychological and somatic symptoms. Results showed that the mindfulness intervention was effective in reducing unhealthy life behaviors, especially for eating (balanced diet, food intake), alcohol consumption, and maladaptive rest habits in a sample of young university students.

Some patterns by gender differences were found in exploratory analyses, where females experienced significant changes in eating habits compared to males. This is an interesting finding if we take into account that eating disorders are more frequent in women, and the fact that social pressure to conform to a normative body image is higher for females. Therefore, the females of this sample may have been more sensitive to change their eating behaviors. Further research with larger samples will be necessary to draw solid conclusions about these patterns.

Maladaptive eating (e.g., eating disorders) frequently persists in adulthood even in the face of significant deterioration in psychological and physiological wellness and overall quality of life [9,52]. It has a strong negative impact on family and social engagement and imposes substantial costs including health-care and social services costs [53] and non-health-care costs associated with obesity such as job or school absenteeism [54], thus increasing the risk of mental disorders and disordered eating development in the future [54,55]. Fortunately, mindful eating programs have proven to be effective in improving body image, reducing eating disorder symptomatology, and preventing weight gain [16,56,57].

Our findings concerning enhancements in rest habits are also consistent with studies showing improvements in the number of hours of sleep and sleep quality after eight weeks of mindfulness training including flow meditation [34,58]. This is a relevant finding since insufficient rest due to bad sleeping habits is related to poor academic performance [2].

Significant differences were also found in the studied variables in the control group, but these were due to a worsening of these behaviors (e.g., substance consumption). We speculate that this result may be due to the proximity of the final exams.

In terms of possible mechanisms of action, in addition to decreased hyperarousal of the sympathetic nervous system and increases in parasympathetic activation through relaxation, mindfulness may have improved healthy life behaviors via the cultivation of new metacognitive resources (e.g., detachment from self-referential processing, and acceptance) that increase perceptual distance from internal and external stimuli [45,46,51]. This is consistent with previous studies of *flow meditation* in which mindfulness was shown to be effective in adolescents with high levels of aggressiveness and lack of impulse control [33]. In addition, mindfulness approaches can intervene by improving the regulation of emotions [16,29] and increasing the experience of positive emotional states [19,59].

The findings of the present study should be considered in light of their limitations, which include (i) a small sample size; (ii) the absence of an active control condition; (iii) reliance on self-report measures; and (iv) lack of follow-up assessment. It would be useful to investigate maintenance effects over a longer period of time. In addition, and given the multifaceted nature of the flow meditation program, it is difficult to know the effect of each of the components in the studied variables.

The current study showed that *flow meditation* was effective for reducing a wide range of unhealthy behaviors, which have mostly been studied separately in samples of students, while developing rest habits. Further research is warranted to replicate these findings as well as to understand the mechanisms that lead to improvements in healthy behaviors following participation in mindfulness training. The potential effects of mindfulness go beyond simply inducing a state of relaxation [17,19]. An additional mechanism of action would be through attitude change [52]. In this sense, over the past decade, the practice of mindfulness has increasingly been linked to sustainable consumption behaviors [52,60]. Schools, universities, and community public health practitioners should provide prevention and intervention efforts among the target population [61]. Given their increasing prevalence of maladaptive health habits among this population, and the associated health-related risks and concurrent psychopathology, greater attention is needed to improve the efficacy of existing treatments. University and college arenas represent an important opportunity for health and nutritional education.

## Figures and Tables

**Figure 1 nutrients-12-02450-f001:**
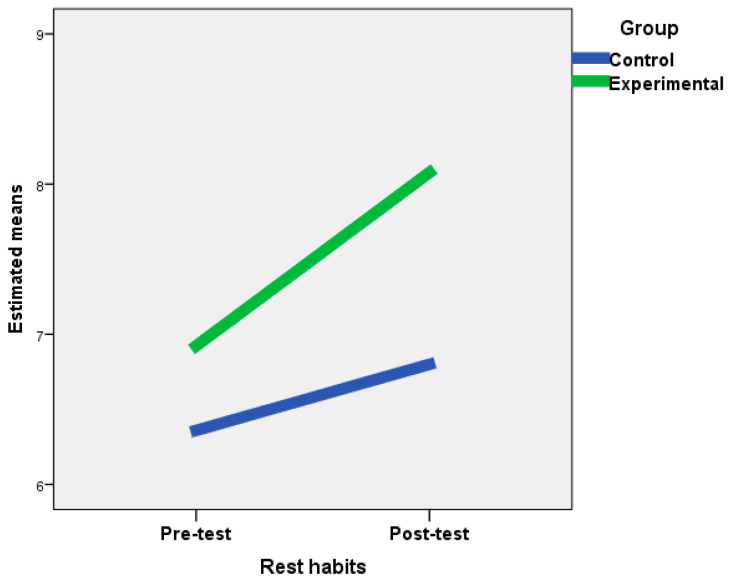
Changes in rest habits for the experimental and control groups after the intervention.

**Figure 2 nutrients-12-02450-f002:**
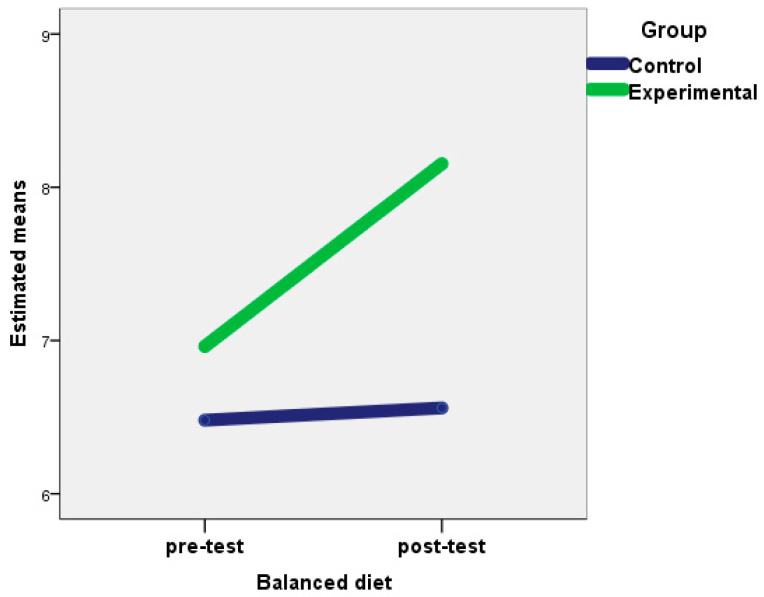
Changes in balanced diet for the experimental and control groups after the intervention.

**Table 1 nutrients-12-02450-t001:** Contents of the flow meditation program by session.

Session 1: Introduction to mindfulness and flow meditation through different Zen metaphors and teachings. Practice of flow meditation. Group discussion.
Session 2: Sense meditation (face). Talk on learning to observe all mental events and let them be and flow; learning the difference between reacting and acting with awareness. Practice of flow meditation (e.g., mindful breathing). Group discussion.
Session 3: Sense meditation (abdomen and chest). Talk on learning how to tolerate negative feelings and thoughts during the practice of mindfulness and in daily life. Practice of flow meditation. Group discussion.
Session 4: Sense meditation (back). Talk on how our minds tend to be in the past or future, and learning how to live in the present moment with awareness. Practice of flow meditation. Group discussion.
Session 5: Sense meditation (arms). Talk on the impermanency of everything. Exercise of counting thoughts and see them constantly flowing in order to be aware of their transitory and impermanency. Additionally, learning how to break the association between thinking, feeling, and acting. Talk on the concept of equanimity and emphasizing that the goal is not learning how to control and dominate our minds, but to learn how not to be overwhelmed and controlled by them. Practice of flow meditation. Group discussion.
Session 6: Sense meditation/Vipassana meditation (all body). Talk on the role of our attitudes in the evaluation of the events and circumstances of our life and how this evaluation determines our life satisfaction and well-being. Learning acceptance of unpleasant states through different exercises. Practice of flow meditation. Group discussion.
Session 7: Sense meditation/Vipassana meditation (all body). Learning how to observe and detect negative mental patterns that prevent us from being happy and satisfied with our lives (e.g., “musts”, “oughts”). Practice of flow meditation. Group discussion. The course ends by encouraging participants to practice mindfulness daily.

**Table 2 nutrients-12-02450-t002:** Descriptive, ANOVA, and Wilcoxon test for related samples: Pre-test and post-test intra-group differences in the study variables and Eta square.

Variables	Experimental							Control						
	Pre-test		Post-Test					Pre-test		Post-Test				
	M	SD	M	SD	Z/F	*p*	r/ *ɳ*^2^*p*	M	DS	M	SD	Z/F	*p*	r/*ɳ*^2^*p*
Tobacco use	6.50	2.98	5.54	2.26	Z = −3.14	**	0.439	6.36	3.22	6.64	3.39	Z = −1.94		0.271
Cannabis consumption	5.54	2.73	4.92	2.07	Z = −3.20	**	0.448	5.08	2.73	5.32	3.06	Z = −2.45	*	0.343
Alcohol consumption	7.38	3.03	6.19	2.11	Z = −3.66	***	0.512	6.60	2.59	7.00	2.81	Z = −1.31		0.183
Meal times	6.96	2.30	7.92	2.27	F = −1.28		0.073	6.80	2.10	7.32	1.99	F = 1.09		0.02
Balanced diet	6.96	1.48	8.15	1.40	F = 7.74	**	0.14	6.48	1.38	6.56	1.55	F = 0.34		0.001
Rest habits	6.92	1.62	8.08	1.32	F = 10.05	**	0.17	6.36	1.57	6.80	1.29	F = 1.39		0.028
Externality	31.73	7.15	27.92	5.78	Z = −4.03	***	0.564	31.56	7.73	31.32	Z = −0.259	8.20		0.362
Decrease in food consumption due to NES	21.85	5.45	18.35	3.03	Z = −3.93	***	0.550	20.92	5.44	21.92	5.09	Z = −2.85	**	0.399
Increase in food consumption due to NES	21.81	6.19	17.62	3.61	Z = −4.12	***	0.577	22.68	5.47	23.36	5.13	Z = −2.70	**	0.378
Food consumption amount	16.27	4.30	12.81	2.51	Z = −4.21	***	0.590	16.52	4.29	17.32	4.17	Z = −3.08	*	0.431
Snacking between meals	3.31	1.12	2.46	0.51	Z = −3.17	**	0.443	3.20	1.04	3.36	0.90	Z = −2.00	*	0.280
Consumption of light products	3.23	1.21	3.38	1.10	F = 3.21		0.062	3.36	0.99	3.16	0.85	F = 5.22	*	0.096
Intake rate	7.08	1.89	6.08	1.06	Z = −2.96	**	0.416	6.92	1.89	7.36	1.65	Z = −2.31	*	0.323

* *p* < 0.05 ** *p* < 0.01 *** *p* ≤ 0.001 Note: NES = negative emotional states. Statistically significant differences were obtained between the two groups in the post-test scores, especially on balanced diet and rest habits as well as in decreasing in food consumption due to NES, increase in food consumption due to NES, amount of consumption, snack between meals, and intake rate (Table 2).

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
