# Peer review of "Promoting a Healthy Lifestyle through Mindfulness in University Students: A Randomized Controlled Trial"

_nutrients, 2020, doi:10.3390/nu12082450_

Round 1

Reviewer 1 Report

Overview:

The actual study conducted seems adequate, but the manuscript needs substantial revision. There are several major issues. First, the authors frame this paper around “habits”, but all they used was a pre/post design to study specific behaviors at two time points. For behavior to truly be considered a habit, it needs to have consistency and automaticity. To call it a habit without this context is dismissing the large psychological literature on habitual behavior. Second, currently the reader has no idea what the measures were in the study because so little description was provided. Third, the results are reported in an unnecessary and confusing way. I have suggested a more simple and clear approach. Additionally, the writing needs significant editing. There were dozens of typos, and the tables and figures were not clear (sometimes missing key information); overall it felt quite sloppy. Finally, the introduction needs to be augmented substantially to provide a more elegant and theory-driven argument for the rationale behind this study. 

Specific Suggestions:

INTRODUCTION:

-The first section of the introduction would benefit from a dramatic revision. There is not enough coherent organization. Also, none of the claims seem theory-driven. The introduction also lacks a description of why these particular health areas were chosen (e.g. why not include exercise?). The introduction would benefit from a more thorough literature review. For example, this line deserves an entire paragraph of explanation: “This change in eating habits is associated with an unhealthy diet and various health-related problems,  and is especially noticeable in university students [1,3].” The emphasis of unhealthy habits on sleep in the introduction feels a bit random. It’s confusing whether lack of sleep is its own unhealthy habit being discussed, or whether you’re trying to emphasize that these other unhealthy habits have repercussions on sleep. Can you draw on national or international recommendations for health to situate your argument?

-Are you interested in clinically significant habits? This introduction seems to blur the lines between unhealthy habits of relatively healthy college students and clinical issues of eating disorders and addictions. It doesn’t seem appropriate to use an umbrella term that encompasses both. 

-In Section 1.1, there needs to be a definition provided of mindfulness-based meditation. Many researchers define it differently. I would recommend focusing this section on how Flow Meditation is taught to avoid confusion. 

-This sentence, “There is increasing literature suggesting that university students may be a population at biological and social risk, often with inadequate health behaviors and life habits” is used in multiple places and is unclear. 

-Statements like this “In fact, at least half of this population presents inadequate health habits” need to be elaborated. 

-You say, “to the authors’ knowledge, this is the first study to specifically investigate the effects of mindfulness on a wide range of variables related to healthy life habits in university students.” I think there are a few papers on this topic. As one example, look up: https://doi.org/10.3389/fnhum.2016.00117

-This is all framed with reference to “habits”, yet there is no description of the literature on habits. This needs to be added.

-There are several typos throughout the paper, particularly with punctuation. 

-This sentence, “If we could identify the factors that determine leading an active or inactive student lifestyle, we would be able to help improve the quality of life in adulthood.” came out of nowhere. Since when is this a study on activity/inactivity?

MATERIALS AND METHODS:

-This study is a pre/post design. How do you claim to study habits? You are studying behavior, but to study a habit, you need ongoing measurement. 

-It seems like the intervention was a hybrid of many teachings. It would be advisable to note this multifaceted nature as a limitation, since it remains unknown which aspect was responsible for the change observed. 

-Was anything related to diet, rest, tobacco, alcohol, or cannabis discussed at any of the intervention group sessions? Please describe either way.

-Example items from the questionnaires for every outcome must be included. As is, the reader has no idea what, for example, “meal times”, “rest habits” or “intake rate”  are. 

RESULTS

-Typo (DS) in Table 1

-Figure 1 is very unprofessional. It looks like a screenshot from an SPSS output.  The labels and parts of the figures themselves are also cut off. For example, in Panel A of Figure 1, only the green line is visible. This must be re-made.

-P values should not be written as .000. Must include the “less than” sign.

-There needs to be a description for why Z is sometimes used and F is sometimes used in Table 2.

-Table 1 includes descriptives, which is good. Table 2 looks unnecessary. You already reported that there were no differences at pre-test, and the differences at post-test are not what we’re most interested in, because these don’t take into consideration the pre-test values. Table 3 also seems somewhat irrelevant. We care most about the interaction of condition x time, and table 3 shows just the effect of time, for each table separately. I suggest you make a single table so the results are clearer for the reader. This table should include 3 panels. The first panel should be the M and SD for the pre-test and post-test of the intervention group, along with the paired t-test for this group across time (t and p values). The second panel should be the M and SD for the pre-test and post-test of the control group, along with the paired t-test for this group across time (t and p values). The third panel should describe the repeated measures ANOVA (F and p values, plus effect size). Then all the relevant data is shown in a single table, which is much more elegant. 

-Were your findings by gender hypothesized ahead of time? This needs to be noted either way. It’s ok if they weren’t, but you need to label these analyses as exploratory. 

-Do you have enough power to reliably run your gender analyses? “statistically significant differences between the pre-test and post-test scores for males in the experimental group in the following variables”. Your sample size was already small to begin with. If you divide this cell further by gender, you’re talking about roughly N=12 per cell. I would delete the entire section on gender from your results section and instead mention in the discussion that you found some interesting patterns by gender in exploratory analyses, but that further research with larger samples will be necessary to make conclusions about these patterns. 

-DISCUSSION

-This is too strong of language “The current study showed that flow meditation was effective for reducing unhealthy habits, 330 including maladaptive eating disorders, addictive behaviors”. Since we know nothing about your measures, it’s hard to say, but I don’t think you were measuring clinical eating disorders or addiction based on the titles of the questionnaires. 

-Works cited section needs lots of formatting corrections

Author Response

PLEASE, SEE ATTACHMENT.

Reviewer 2 Report

General comments

This article presents a randomized controlled study on the effects of a second generation mindfulness-based program on student health behaviors and lifestyles. Although the sample size is small, the results are promising. However, a major limitation is the inactive control group as student adjustment to higher education is influenced by social support, without an active control group it is not possible to determine whether the effect was due to mindfulness-based practices or to the social support and social connectedness the students benefited from through this group.

The originality and significance of this study relies on the fact that this intervention is relatively new and the results are promising. Furthermore, the authors measured a number of health behaviors which have mostly been studied separately in student samples mindfulness-based programs. The results are well presented and the discussion is based on the results obtained and presents the limitations. A few typos need to be corrected.  

Detailed comments

In the abstract: by “emotional states negative” should read “negative emotional states”

Practice time controlled for? Did the women practice more? = confounding variable?

L76, L89, L103, L104, L109, end of sentence add full stop.

L109: students “who” rather than “that”

P6 part of the figure does not appear

It would be useful to explain why you chose a flow mindfulness program. As it is a relatively new program, it may be useful to add a full outline of the program in order to be able to see what types of practices are proposed. It could be added as online supplementary materials.

Also, it was unclear why in preceding articles on this program there were 10 sessions and here only 7. Was it adapted more specifically to this population and what was different from the other Flow meditation programs here?

In the sampling, were all the students recruited in both groups motivated to take part in a mindfulness-based program? To what extent did they practice outside the sessions and what were the suggested practices? (could also appear in the Supplementary Materials section).

Author Response

PLEASE, SEE ATTACHMENT.

Round 2

Reviewer 1 Report

This is a much improved version of the manuscript. They did a nice job with Table 2, which is now easier to understand. I would suggest further adjusting the figures which look like screenshots. For example, the small font size is quite hard to read.